# Genome-Wide Identification of the Rose *SWEET* Gene Family and Their Different Expression Profiles in Cold Response between Two Rose Species

**DOI:** 10.3390/plants12071474

**Published:** 2023-03-28

**Authors:** Xiangshang Song, Yaping Kou, Mingao Duan, Bo Feng, Xiaoyun Yu, Ruidong Jia, Xin Zhao, Hong Ge, Shuhua Yang

**Affiliations:** State Key Laboratory of Vegetable Biobreeding, Key Laboratory of Biology and Genetic Improvement of Flower Crops (North China), Ministry of Agriculture and Rural Affairs, Institute of Vegetables and Flowers, Chinese Academy of Agricultural Sciences, Beijing 100081, China

**Keywords:** rose, *SWEET* gene family, sugar transport, gene expression, cold response

## Abstract

Sugars Will Eventually be Exported Transporter (SWEET) gene family plays indispensable roles in plant physiological activities, development processes, and responses to biotic and abiotic stresses, but no information is known for roses. In this study, a total of 25 *RcSWEET* genes were identified in *Rosa chinensis* ‘Old Blush’ by genome-wide analysis and clustered into four subgroups based on their phylogenetic relationships. The genomic features, including gene structures, conserved motifs, and gene duplication among the chromosomes of *RcSWEET* genes, were characterized. Seventeen types of *cis*-acting elements among the *RcSWEET* genes were predicted to exhibit their potential regulatory roles during biotic and abiotic stress and hormone responses. Tissue-specific and cold-response expression profiles based on transcriptome data showed that *SWEETs* play widely varying roles in development and stress tolerance in two rose species. Moreover, the different expression patterns of cold-response *SWEET* genes were verified by qRT-PCR between the moderately cold-resistant species *R. chinensis* ‘Old Blush’ and the extremely cold-resistant species *R. beggeriana*. Especially, *SWEET2a* and *SWEET10c* exhibited species differences after cold treatment and were sharply upregulated in the leaves of *R. beggeriana* but not *R. chinensis* ‘Old Blush’, indicating that these two genes may be the crucial candidates that participate in cold tolerance in *R. beggeriana*. Our results provide the foundation for function analysis of the *SWEET* gene family in roses, and will contribute to the breeding of cold-tolerant varieties of roses.

## 1. Introduction

Sugars Will Eventually be Exported Transporter (SWEET) is a new type of sugar transporter that was first identified in *Arabidopsis thaliana* in 2010 [1]. Comparing with monosaccharide transporters (MST) and sucrose transporters (SUT), SWEET proteins are capable of transporting sugar bidirectionally without energy dependence, and the types of substrates for sugar transport are more extensive [2]. In plants, SWEET proteins usually contain seven transmembrane helices (TMHs); two MtN3/saliva domains with three TMHs are connected by the fourth TMH to form a “3-1-3” structure [3]. The SWEET family is divided into four Clades. Different clades have different preferences for monosaccharides or disaccharides, in which Clades I and II prefer transporting hexose and Clade III prefer transporting sucrose, while SWEET proteins of Clade IV tend to transport fructose on the tonoplast [4,5,6,7].

Due to the advances in whole-genome sequencing in plants, the genome-wide identification of *SWEET* genes has been reported in many crops, vegetables, and fruits, such as rice (*Oryza sativa*) [8], soybean (*Glycine max*) [9], sorghum (*Sorghum bicolor*) [10], tomato (*Solanum lycopersicum*) [11], potato (*Solanum tuberosum*) [12], cabbage (*Brassica oleracea*) [13], tea plant (*Camellia sinensis*) [14], apple (*Malus domestica*) [15], grape (*Vitis vinifera*) [16], and banana (*Musa acuminate*) [17]. Previous studies show that *SWEET* genes in plants are involved in many biological processes, including the regulation of pollen development, nectar secretion, seed development, phloem loading, and leaf senescence. In Arabidopsis, loss of function of *AtSWEET8/RPG1* led to male sterility, indicating a key role in the maintenance of pollen viability [1,18]. *AtSWEET13* is also demonstrated to have the function for pollen development in Arabidopsis [19]. *AtSWEET9/Nec1* and its homologous genes in tobacco (*Nicotiana attenuata*) and rape (*Brassica rapa*) are proven to be essential for nectar secretion [20]. AtSWEET10 is reported to be the downstream of FLOWERING LOCUS T (FT) during floral transition in Arabidopsis [21]. AtSWEET11, 12, and 15 are beneficial to seed filling by mediating sucrose transfer from the seed coat to the endosperm [22]. AtSWEET11 and 12 are also responsible for phloem loading for long-distance transport of sucrose [7,23]. OsSWEET4, 11, and 15 in rice (*Oryza sativa*) and ZmSWEET4 in maize (*Zea mays*) are involved in the transport of hexose or sucrose in the endosperm to promote seed filling [24,25]. AtSWEET13 and 14 may be involved in modulating the GA response in Arabidopsis [26]. The NAC transcription factor ORE1 could bind to the *AtSWEET15* promoter to positively regulate leaf senescence [27]. CmSWEET17 could be involved in the process of sucrose-induced axillary bud outgrowth in chrysanthemum (*Chrysanthemum moriflorum*), possibly via the auxin transport pathway [28].

*SWEET* genes also play very important roles in response to biotic and abiotic stresses. Overexpression of *AtSWEET16* and its homologous genes in tea and apples can enhance the cold resistance of the transgenic calli or plants [14,29,30]. Moreover, overexpression of *CsSWEET1a* and *CsSWEET17* from tea also improves the cold tolerance of the transgenic Arabidopsis plants [31]. Loss of function in *AtSWEET17* may affect lateral root development and lead to impaired drought resistance [32]. *AtSWEET2* inhibits *Pythium* infection by reducing the availability of sugars in the rhizosphere of Arabidopsis seedlings [33,34]. In grape, overexpression of the *VvSWEET4* increases hexose content in hair roots and enhances resistance to *Pythium* [35]. In rice, *OsSWEET11*, *13*, and *14* are involved in resistance to bacterial blight disease by regulation of the upstream transcription factors [36,37,38]. However, no information about the *SWEET* gene family is known for the genus *Rosa* L.

Rose is one of the most popular flowers in the world and is widely used in cut flowers, potted flowers, garden cultivation, and essential oil production [39]. Cold stress affects the growth and distribution of plants, resulting in freezing injuries and even the death of plants [40,41,42]. The rose industry suffers great commercial losses every year due to cold stress [40,41,42]. *Rosa beggeriana* is a wild species that originated in cold and arid Central Asia, including northwest China. It has extreme cold resistance, which can be inherited through interspecific hybridization with modern rose cultivars [43]. Our previous study found that the significantly higher freezing tolerance in the shoot of *R. beggeriana* than that of *R. fortuneana*, which originated in southeastern China, might be due to the stronger ability of the spatial transfer of soluble sugars from leaves to shoots and consequently the storage of soluble sugars as starch in the shoots during overwintering [44]. The transcriptome analyses in *R. multiflora* and *R. xanthina* further indicated that the pathways of starch and sucrose metabolism were activated during cold stress [42,45]. However, it is not clear whether and which members of the SWEET gene family may participate in the cold response of a rose. Furthermore, it is not sure if there are species differences in these cold-response SWEET genes among the cold-sensitive and cold-tolerant species in *Rosa* L.

In this study, the *SWEET* family genes in rose were identified, and their gene structures, motif compositions, phylogenetic relationships, chromosome locations, gene duplications, and *cis*-acting elements were analyzed on the basis of the genome sequence of *R. chinensis* ‘Old blush’. The comparison of expression patterns of the *SWEET* family genes was further investigated between *R. chinensis* ‘Old blush’ and *R. beggeriana* during cold treatment. Finally, the analysis of subcellular localization was conducted on the screened cold-response *RbSWEET* genes in *R. beggeriana*. Our results will contribute to a better understanding of the *SWEET* gene family in rose and provide valuable information for further functional analysis of *RbSWEET* genes in response to cold tolerance.

## 2. Results

### 2.1. Genome-Wide Identification of the SWEET Family Genes in R. chinensis

A total of 29 SWEET proteins were identified on the basis of HMM profiles of the SWEET domain (PF03083) search results. Meanwhile, 28 SWEET homologous proteins were obtained by using 17 protein sequences of AtSWEETs to BLASTP in the *R. chinensis* proteome with an e-value threshold of 1 × 10^−5^. After the intersection of these two results, 25 *SWEET* genes were finally identified in *R. chinensis* according to the conserved domains in the relevant database (Table 1). These genes were named from *RcSWEET1* to *RcSWEET17b* according to their homology with *A. thaliana*. Protein characterization analysis showed that the ranges of genomic DNA (gDNA) and coding sequences (CDS) are 717~3289 bp and 708~933 bp, respectively. The amino acid length of RcSWEET family members ranged from 235 (RcSWEET2a, RcSWEET2b, and RcSWEET5c) to 310 (RcSWEET12), and the average molecular weight (MW) ranged from 26,284.25 (RcSWEET2b) to 34,702.73 (RcSWEET12). The range of the theoretical isoelectric point (pI) is between 5.93 (RcSWEET15b) and 10.11 (RcSWEET10c). The TMHMM predicted results showed that all 25 RcSWEET proteins contained 7 transmembrane domains, indicating that the transmembrane structure of RcSWEET proteins was relatively conservative (Appendix A).

### 2.2. Gene Structure, Motif, and Phylogenetic Analysis of the RcSWEET Genes

The gene structures, including untranslated regions (UTRs), exons, and introns, of each *RcSWEET* gene were predicted based on gDNA and CDS sequences (Appendix A). The majority of the *RcSWEET* genes (*RcSWEET1*, *RcSWEET2a*, *RcSWEET2b*, *RcSWEET3*, *RcSWEET4a*, *RcSWEET5a*, *RcSWEET5c*, *RcSWEET5d*, *RcSWEET9*, *RcSWEET10a*, *RcSWEET10b*, *RcSWEET10c*, *RcSWEET11a*, *RcSWEET12*, *RcSWEET15a*, *RcSWEET15b*, *RcSWEET17a*, and *RcSWEET17b*) contained six exons. Two *RcSWEET* genes (*RcSWEET4b* and *RcSWEET11b*) had five exons, whereas five *RcSWEET* genes (*RcSWEET5b*, *RcSWEET5e*, *RcSWEET5f*, *RcSWEET5g*, and *RcSWEET5h*) had only one exon and no intron (Figure 1B). The conserved motifs were investigated based on the protein sequences of the *RcSWEET* genes. Ten consensus motifs were detected in the *RcSWEET* genes (Figure 1C). Each of the 25 members contained two MtN3_slv domains that were composed of three transmembrane helical domains. To investigate the phylogenetic relationship of the *SWEET* genes in *R. chinensis*, the identified *SWEET* family members of rose were constructed alone and combined with the *SWEET* family members of rice and Arabidopsis. The results showed that 25 *RcSWEETs* were divided into four clades, including 4 members in Clade I, 10 members in Clade II, 9 members in Clade III, and 2 members in Clade IV (Figure 1A and Figure 2).

### 2.3. Chromosomal Location and Gene Duplication Analysis of the RcSWEET Genes

The analysis of chromosomal location showed that 25 *RcSWEET* genes were distributed on six chromosomes of *R. chinensis* ‘Old Blush’. Chr6 contained the greatest number of six *RcSWEET* genes, including *RcSWEET1*, *RcSWEET9*, *RcSWEET10b*, *RcSWEET10c*, *RcSWEET11b*, and *RcSWEET11b*. Both Chr1 and Chr7 contained five genes, which were *RcSWEET3, RcSWEET5f, RcSWEET5g, RcSWEET5h*, and *RcSWEET17a* on Chr1 and *RcSWEET4b, RcSWEET5a, RcSWEET5d, RcSWEET10a*, and *RcSWEET12* on Chr7. There were four genes, including *RcSWEET5e, RcSWEET15a, RcSWEET15b*, and *RcSWEET17b*, on Chr4. Three genes, including *RcSWEET2a, RcSWEET2b*, and *RcSWEET5b*, are located on Chr5. Only two genes, *RcSWEET4a* and *RcSWEET5c*, are distributed on Chr2. However, there was no *RcSWEET* gene on Chr3.

Gene duplication events usually play an indispensable role in gene expansion and the evolution of gene families. A pair of tandemly duplicated genes (*RcSWEET10a* and *RcSWEET12*) located on Chr7, and another cluster of five tandemly duplicated genes (*RcSWEET9*, *RcSWEET10b*, *RcSWEET10c*, *RcSWEET11a*, and *RcSWEET11b*) located on Chr6 (Figure 3A). In addition, two segmental duplication events, including *RcSWEET4a*/*RcSWEET5a* and *RcSWEET12*/*RcSWEET15a*, were detected on three different chromosomes (Figure 3B). The selection pressure was further estimated by the Ka (nonsynonymous) and Ks (synonymous) substitution ratios [46]. Only one pair of segmentally duplicated *RcSWEETs*, namely *RcSWEET4a* and *RcSWEET5a*, had a Ka/Ks ratio of 0.15, which was significant, and indicated a synonymous change that has been selected during plant genome evolution (Appendix A).

### 2.4. Cis-Elements Analysis in the Promoters of the RcSWEET Genes

To further study the potential regulatory mechanisms of the *RcSWEET* genes in response to abiotic stress, the 2000 bp upstream sequences from the translation start sites of the *RcSWEET* genes were submitted into PlantCARE to detect the cis-elements. Seventeen cis-acting elements associated with stress and hormones were identified, including ABRE, ARE, AuxRR core, Box 4, CGTCA-motif, circadian, ERE, GARE-motif, LTR, MBS, P-box, TATC-box, TCA-element, TC-rich, TGACG-motif, TGA-element, and WUN-motif (Figure 4). The details of the cis-acting elements contained in each *RcSWEET* gene have been listed in Appendix A. All the *RcSWEET* genes possessed at least one stress-relevant cis-element; ABRE, ARE, and Box 4 were the most abundant genes in the *SWEET* family, indicating that abscisic acid, light, and anaerobic conditions had a great influence on the expression of *RcSWEET* genes, and these cis-acting elements were also related to stress response. In particular, 13 elements (52%) contained one or more LTR, suggesting a potential cold-stress response under low temperature conditions.

### 2.5. Expression Patterns of RcSWEET Genes in Different Tissues

The spatial expression patterns of *RcSWEET* genes in different tissues have been analyzed according to the published transcriptome sequencing (RNA-seq) data in *R. chinensis* ‘Old Blush’ under the growth condition of a 12 h (25 °C) day/12 h (18 °C) night cycle [47]. There were 13 *RcSWEET* genes (52%) expressed in roots, 10 genes (40%) in stems, 9 genes (36%) in prickles and leaves, 18 genes (72%) in stamens, and 15 genes (60%) in pistils and ovaries (Figure 5). Six *RcSWEET* genes, including *RcSWEET1*, *RcSWEET2a*, *RcSWEET2b*, *RcSWEET4b*, *RcSWEET5h*, and *RcSWEET17a*, were expressed in all tissues. Especially, *RcSWEET1* was highly expressed (FPKM > 20) in prickles, leaves, stamens, pistils, and ovaries. *RcSWEET4b* was highly expressed (FPKM > 20) in roots, stems, prickles, stamens, pistils, and ovaries. Some of the *RcSWEET* genes exhibited similar expression patterns in the same tissues. For example, *RcSWEET5d* and *RcSWEET10a* showed extremely high expression levels (FPKM > 95) in stamen. Moreover, the highly expressed levels (FPKM > 20) were presented in *RcSWEET2a* and *RcSWEET3* in leaves, as well as *RcSWEET12* and *RcSWEET15a* in roots.

### 2.6. Expression Patterns of Cold-Response SWEET Genes in Two Rosa Species

Identically as in *R. chinensis*, 25 *RbSWEET* genes have been identified in *R. beggeriana* based on the transcriptome analysis of leaves and shoots in response to cold stress (Figure 6). At least 12 genes were statistically detected under the growth temperature of 23/18 °C, including *RbSWEET1*, *RbSWEET2a*, *RbSWEET2b*, *RbSWEET3*, *RbSWEET4b*, *RbSWEET10c*, *RbSWEET11a*, *RbSWEET15a*, *RbSWEET17a*, and *RbSWEET17b* in both tissues, as well as *RbSWEET5f* and *RbSWEET5h* in shoots. The above results presented very similar expression patterns in the leaves and shoots of *R. chinensis* ‘Old Blush’. After cold stress at 4 °C, the differentially expressed genes (DEGs) were screened out of the leaves and shoots of *R. beggeriana*. In leaves, *RbSWEET2a* and *RbSWEET10c* were upregulated, while *RbSWEET4b* was downregulated during cold stress. *RbSWEET1* and *RbSWEET15a* were downregulated at 1 h and then upregulated until 24 h of cold stress. On the contrary, *RbSWEET3* and *RbSWEET2b* presented the upregulated expression level at 1 h and then the downregulated expression level at 6 h and 24 h during cold stress. In shoots, *RbSWEET10c*, *RbSWEET4b*, *RbSWEET1*, and *RbSWEET2b* exhibited very similar expression trends as those genes in leaves. In contrast to leaves, there were much higher expression levels of *RbSWEET10c* and *RbSWEET4b* in shoots. *RbSWEET2a* and *RbSWEET15a* exhibited the downregulated expression levels, but *RbSWEET3* fluctuated in shoots during cold stress. In addition, *RbSWEET17a* did not show a different effect on growth temperature but maintained the relatively higher expression levels in leaves and shoots under both growth temperatures.

The expression patterns of DEGs or highly expressed *SWEET* genes in leaves and shoots of *R. chinensis* ‘Old Blush’ and *R. beggeriana* during cold treatment are shown in Figure 7. In the leaves of *R. chinensis* ‘Old Blush’, significantly increased relative expression levels were found in *RcSWEET1* at 6 h and *RcSWEET15a* at 24 h during cold stress. *RcSWEET2a* and *RcSWEET10c* showed a significant decline at 1 h but recovered at 6 h and 24 h. *RcSWEET2b* did not significantly change during cold stress. The relative expression levels of *RcSWEET3* and *RcSWEET17a* fluctuated during cold stress, significantly decreasing at 1 h, increasing at 6 h, and declining at 24 h. *RcSWEET4b* exhibited the exact opposite trend as *RcSWEET3*, but maintained significantly higher relative expression levels during cold treatment than those at 0 h. In shoots of *R. chinensis* ‘Old Blush’, *RcSWEET1* was significantly upregulated at 1 h and 24 h, but *RcSWEET2a* was unchanged during cold treatment. The increased expression levels were observed in *RcSWEET2b* at 6 h and in *RcSWEET10c* at 1 h. *RcSWEET3* was dramatically upregulated at 1 h, then significantly downregulated at 6 h, but recovered at 24 h. The relative expression levels of *RcSWEET4b*, *RcSWEET15a*, and *RcSWEET17a* significantly increased at 1 h and 6 h but sharply declined at 24 h.

As for the leaves and shoots of *R. beggeriana*, the expression patterns of most *RbSWEETs* determined using qRT-PCR detection were similar to the transcriptome results. However, the qRT-PCR analysis showed the expression levels of *RbSWEET4b* increased at 1 h and 24 h in leaves and enhanced at 1 h and 6 h in shoots, instead of the decreased results in the RNA-seq data. In addition, *RbSWEET17a* did not alter in the transcriptome data but significantly declined in leaves by qRT-PCR analysis during cold stress.

The further comparison analysis disclosed that *SWEET* genes showed various expression patterns in two species during cold treatment (Figure 7). For instance, *SWEET1* and *SWEET4b* exhibited upregulated trends in leaves and shoots of both species. The expression patterns of *SWEET2a* and *SWEET10c* were quite similar in the shoots of both species. In leaves, *RcSWEET2a* and *RcSWEET10c* sharply declined at 1 h, then recovered to the same expression levels as those genes at 0 h, while *RbSWEET2a* and *RbSWEET10c* stably enhanced and reached the significant differences after 6 h and 24 h of cold treatment, respectively. Moreover, *SWEET2b*, *SWEET3*, and *SWEET15a* exhibited upregulated trends in the shoots of *R. chinensis* ‘Old Blush’ at the beginning of cold stress, but downregulated in the shoots of *R. beggeriana* during the whole cold treatment.

### 2.7. Subcellular Localization of Three RbSWEET Genes

Based on the expression patterns of the *SWEET* genes in both species, three *RbSWEET* genes were selected for subcellular localization. *RbSWEET1*, *RbSWEET2a*, and *RbSWEET10c* were transiently overexpressed in tobacco leaf epidermal cells. As shown in Figure 8, *RbSWEET1* and *RbSWEET10c* were found to be localized to the plasma membrane, and *RbSWEET2a* was found to be localized to the tonoplast, whereas the control (GFP alone) was observed clearly in the plasma membrane, cytoplasm, and nucleus.

## 3. Discussion

The SWEET gene family has been identified in many major crops such as rice [8], sorghum [10], and soybean [9], as well as important horticultural plants such as cabbage [13], potato [12], tomato [11], and apple [15]. However, little information has been reported on roses and other ornamental plants, especially their molecular regulatory mechanisms in cold response. The genome sequencing of *R. chinensis* ‘Old Blush’ laid the foundation for the whole genome identification and analysis of the rose gene family [48]. In this study, 25 *RcSWEET* genes were identified and classified into four subfamilies (Clades I–IV) according to their phylogenetic evolutionary relationship, including four members in Clade I, ten members in Clade II, nine members in Clade III, and two members in Clade IV (Figure 1A). The phylogenetic tree constructed with the SWEET family proteins of Arabidopsis, rice, and rose showed that the SWEET proteins of rose were more closely related to Arabidopsis, which also belongs to dicotyledons (Figure 2). Analysis of the *RcSWEET* gene structure indicated that most genes contained five or six exons, which was consistent with the analysis results in other species such as cabbage [13] and banana [17]. Moreover, similar intron and extron arrangements were found in the same subfamily members (Figure 1B), which implied a different function in each clade. In plants, SWEET proteins usually consist of seven transmembrane helices (TMHs), and two MtN3/saliva domains with three TMHs connected by the fourth transmembrane helix to form a “3-1-3” structure [3]. Compared with plants and other eukaryotes, SWEET protein in prokaryotes usually has only three TMHs, which is named SemiSWEET protein, suggesting that eukaryotic SWEET protein is likely to evolve due to gene replication or fusion [6]. Ten consensus motifs were detected in the RcSWEET proteins, and all the RcSWEET proteins contained two MtN3_slv domains. This result suggested that the structure of RcSWEET proteins was highly conserved (Figure 1C).

Gene duplication events, including whole-genome duplication (WGD), tandem duplication, and segmental duplication events, usually result in functional segregation during gene expansion and evolution [46]. A total of 11 *RcSWEET* genes were identified as being tandemly or segmentally duplicated, which included a pair of tandemly duplicated genes (*RcSWEET10a* and *RcSWEET12*) located on Chr7, another cluster of five tandemly duplicated genes (*RcSWEET9*, *RcSWEET10b*, *RcSWEET10c*, *RcSWEET11a*, and *RcSWEET11b*) located on Chr6, and two pairs of genes (*RcSWEET4a-RcSWEET5a* and *RcSWEET12-RcSWEET15a*) were segmentally duplicated (Figure 3). These results indicated that tandem duplication and segmental duplication synergistically contribute to the expansion of the *RcSWEET* gene family, but the former is more influential than the latter. Although the duplicated *RcSWEET* genes can be derived from a common ancestor, their expression patterns and functions may have diverged. For example, the expression levels of the tandemly duplicated gene pair of *RcSWEET10a*-*RcSWEET12* identified in this work were very different in the various tissues of *R. chinensis* ‘Old Blush’. In Arabidopsis, the unique *AtSWEET10* functions for early flowering [21], but there are three *RcSWEET10a/b/c* genes in rose. Moreover, *RcSWEET10a* exhibited higher expression levels in flowers, pistils, and ovaries, suggesting that *RcSWEET10a* may play an important role during the reproductive development. *AtSWEET12* was reported to be responsible for phloem loading and seed filling in Arabidopsis [11,25,28]. However, *RcSWEET12* was highly expressed in roots, suggesting that it might be involved in root development in roses.

The regulation of gene expression in higher plants usually occurs at the transcriptional level, which is coordinated by various *cis*-acting elements and trans-acting factors [49]. However, there are few studies on *SWEET* transcriptional regulation in plants. For example, the NAC transcription factor ORE1 and three abscisic acid (ABA)-responsive element (ABRE)-binding transcription factors, ABF2 (AREB1), ABF3, and ABF4 (AREB2), were found to be involved in directly regulating senescence-associated genes *AtSWEET15* by binding to their promoters in Arabidopsis [27,50]. A waterlogging-responsive ERF (MaRAP2-4) from mint (*Mentha spp.*) was reported to specifically target the DRE or GCC box in the *AtSWEEET10* promoter to regulate soluble sugar availability and waterlogging tolerance [51]. The rice transcription factor OsDOF1 is involved in the regulation of pathogen interactions by targeting the promoters of *SWEET11/14.* In this study, a wide variety of *cis*-acting elements have been identified in the promoters of *RcSWEET* genes, including *cis*-acting elements associated with stress and hormone response, suggesting that family members are involved in complex signaling pathway regulation (Figure 4). The number of ABRE elements was the largest in the whole family. Previous studies showed that the DREB transcription factor could bind to ABRE in the gene promoter region to respond to cold stress, suggesting that *RcSWEETs* containing these *cis* elements might be involved in the regulation of the cold response in roses [52]. Moreover, 13 *RcSWEETs* contained LTR elements, which might directly participate in the regulation of low temperature response.

Low temperature is one of the typical abiotic stress factors for plants, which has an important effect on the growth and development of plants and their geographical distribution [40]. Cold acclimation at non-freezing temperatures can enhance cold resistance and induce many physiological and biochemical changes in plants, such as the accumulation of osmotic regulatory substances, the removal of reactive oxygen species, and the expression of cold responsive genes (CORs) [53,54,55,56]. It is true that the cold-induced increase of the soluble sugar content should be directly associated with the redistribution and balance of sugar in plants through the regulation of sugar transporters [57]. Previous studies showed that *SWEET* genes could be induced by low temperatures and participate in cold stress responses in many plants. For example, the overexpression of *DlSWEET1* in the tropic fruit plant longan (*Dimocarpus longan*) improved cold tolerance in transgenic *Arabidopsis* plants [58]. *BoSWEET2b*, *BoSWEET4a*, and *BoSWEET15b* were significantly upregulated in the cabbage leaves after cold stress [13]. A hexose transporter, CsSWEET2, from cucumber (*Cucumis sativus*), can improve cold tolerance in *Arabidopsis* [59]. The double mutant of *sweet11 sweet12* exhibited significantly increased freezing tolerance in the leaves of Arabidopsis, indicating that AtSWEET11 and AtSWEET12 function negatively under cold stress [60]. The Arabidopsis transgenic plants with the overexpressed *AtSWEET16* gene exhibited higher cold resistance than wild type [29]. Moreover, overexpression of *MdSWEET16* in apple ‘Orin’ calli was able to increase their cold tolerance compared with *MdSWEET16* RNA interference calli [30]. Enhancing the expression of CsSWEET16/1a/17 in tea plants could improve cold tolerance in the transgenic Arabidopsis plants [14,31]. In addition, more than 90% of *MaSWEETs* in banana leaves were induced in response to cold stress in two different varieties [17].

In the present study, all eight candidate genes statistically presented significant differences in the leaves or shoots of two rose species during the different hours of cold treatment. Many of the SWEET genes that have been proven to be activated under cold stress in other species were also found in the two rose species studied. For instance, *SWEET1* has been recently reported to play an important role in cold resistance in longan [57], which is also enhanced in two rose species after cold stress. There were increased expression levels of the homologous genes *CsSWEET2* in cucumber [59], *BoSWEET2b* in cabbage [13], and *RbSWEET2a* in rose after cold treatment. Moreover, *BoSWEET4a* and *BoSWEET15b* in cabbage [13], as well as *RcSWEET4b* and *RcSWEET15a* in rose, could be activated under the cold growth temperature. We also found that the expression level of *SWEET10c* was significantly enhanced in the tissues of rose species. As far as we know, this is the first report that the *SWEET10* gene could be induced under low temperature in plants. In Arabidopsis, *AtSWEET10* acted downstream of FT to promote plant flowering [21]. *PbSWEET10* might also contribute to pollen development in the Chinese white pear [61]. Moreover, *StSWEET10b* was downregulated in the leaves of potatoes after drought treatment [62]. Recently, *GmSWEET10a/b* were reported to participate in the domestication of seed development in soybean [63].

Analysis using qRT-PCR further demonstrated that the expression levels of *SWEET* genes varied between the two species during cold treatment. The similar upregulated expression patterns of *SWEET1* and *SWEET4b* were observed in leaves and shoots of both species, implying that these two genes should be relatively conserved for the cold response in the genus *Rosa* L. Since *RbSWEET1* was localized in the plasma membrane, it may function on the cellular level for the accumulation of soluble sugars after cold stress. *RbSWEET2a* and *RbSWEET10c* were significantly upregulated in the leaves of *R. beggeriana*, while *RcSWEET2a* and *RcSWEET10c* sharply declined at 1 h and then recovered in the leaves of *R. chinensis* ‘Old Blush’ after cold treatment. The results suggested that *RcSWEET2a* and *RcSWEET10c* could be the crucial candidate genes for the cold-induced transportation of soluble sugars in the leaves of the deciduous and extremely cold-tolerant species *R. beggeriana*. As for *R. chinensis* ‘Old Blush’, *RcSWEET2a* and *RcSWEET10c* might be more sensitive to cold stress than their homologues in *R. beggeriana*, where low temperatures could initially impair their activities but then recover after the cold adaptation in this evergreen and moderately cold-tolerant species. Moreover, RcSWEET2a was localized in the tonoplast membrane, but RcSWEET10c was localized in the plasma membrane, implying that *RbSWEET2a* and *RbSWEET10c* may have different functions for the acquisition of cold tolerance in *R. beggeriana*.

## 4. Materials and Methods

### 4.1. Identification of the SWEET Family Genes in R. chinensis

The genome sequence and annotation of *R. chinensis* ‘Old Blush’ were downloaded from the NCBI genome database (https://www.ncbi.nlm.nih.gov/genome/?term=rosa+chinensis, accessed on 18 December 2022) [48]. The Hidden Markov Model (HMM) profiles of the SWEET domain (PF03083) were downloaded from the Pfam database (http://pfam.xfam.org/, accessed on 18 December 2022) and used to search the SWEET proteins of *R. chinensis* with HMMER (3.3.2) software (e-value < 0.01) [64]. In addition, the protein sequences of 17 AtSWEETs [1] were downloaded from TAIR (http://www.arabidopsis.org/, accessed on 18 December 2022) and used as queries to search the *R. chinensis* proteome by using BLASTP with an e-value threshold of 1 × 10^−5^. Then the results of the two methods are intersected to obtain the candidate sequences, whose conserved domains were further identified by the Pfam database (https://pfam.xfam.org/search#tabview=tab1, accessed on 18 December 2022) and the Conserved Domain Database (http://www.ncbi.nlm.nih.gov/Structure/cdd/wrpsb.cgi/, accessed on 18 December 2022). Finally, the ProtParam tool (http://web.expasy.org/prot-param, accessed on 18 December 2022) was used to analyze the sequence length, molecular weight, and theoretical isoelectric point values of each RcSWEET protein. The transmembrane helix of RcSWEET proteins was predicted by the TMHMM Server version 2.0 (http://www.cbs.dtu.dk/services/TMHMM/, accessed on 18 December 2022) [65].

### 4.2. Gene Structure, Motif, and Phylogenetic Analysis

Based on the genomic sequence and coding sequence, the gene structure of *RcSWEET* genes was predicted by the Gene Structure Display Server (GSDS; http://gsds.gao-lab.org/, accessed on 18 December 2022) [66]. The conserved motifs in full-length SWEET proteins were identified by MEME (https://meme-suite.org/meme/tools/meme, accessed on 18 December 2022) [67]. Multiple sequence alignments were carried out considering the full-length SWEET protein sequences from *A. thaliana, O. sativa*, and *R. chinensis* (Appendix A). Subsequently, an unrooted neighbor-joining (NJ) tree was constructed by MEGA 7.0 with 1000 bootstrap replications [68].

### 4.3. Chromosomal Location and Tandem Duplication Analysis

The information on location for each *RcSWEET* gene was obtained from the GFF genome annotation of *R. chinensis* ‘Old Blush’. The chromosomal localization of the *RcSWEET* genes was mapped using MapChart (2.3.2) software [69]. The MCscanX software was used to search for tandem and duplicated genes. The Ka and Ks values for duplicated gene pairs (Appendix A) were calculated based on the coding sequence alignments using the Ka/Ks calculator in TBtools [46].

### 4.4. Cis-Acting Element Analysis

The upstream sequences (2000 bp) of the 25 *RcSWEET* coding sequences were retrieved and then submitted to PlantCARE (http://bioinformatics.psb.ugent.be/webtools/plantcare/html/, accessed on 18 December 2022) to identify cis-acting elements [70]. The prediction results of cis-acting elements were illustrated by GSDS.

### 4.5. RNA-Seq Data Analysis

To investigate the expression patterns of *RcSWEET* genes in different tissues of *R. chinensis* ‘Old Blush’, organ specificity transcriptome data (PRJNA546486) of the rose, including roots, stems, leaves, stamens, prickles, pistils, and ovaries, was extracted from the NCBI SRA database (https://www.ncbi.nlm.nih.gov/sra/, accessed on 18 December 2022) [47]. The raw data have been listed in Appendix A. FPKM (Fragments Per Kilobase of transcript per Million mapped reads) was used to examine the expression levels, and the heatmap of *RcSWEET* genes in different tissues was generated using TBtools based on the log_2_FPKM values.

The grafted seedlings of *R. beggeriana* ‘Old Blush’ with the rootstock of *R. multiflora* were placed in a growth chamber with an air temperature of 23 °C day/18 °C night and a photoperiod of 10 h light/14 h dark. After a week of acclimatization, the grafted seedlings were maintained at 4 °C for 0, 1, 6, and 24 h. Current-year shoots of similar thickness and intact leaves were selected as sampling materials, and all the samples were collected in three biological replicates (the data has not been published yet). The Illumina HiSeq2500 high-throughput sequencing platform was used to sequence the cDNA library and obtain the raw data. All high-quality reads of each sample that passed quality control were mapped to the *R. chinensis* ‘Old Blush’ RchiOBHm-V2 genome. Gene expression levels were calculated as FPKM. The *RbSWEET* genes have been identified, and the transcriptome data are listed in Appendix A. The heatmap of *RbSWEET* gene expression profiles was generated using TBtools based on the log_2_FPKM values. DEGs were screened according to certain criteria (fold change ≥ 2 and *p*-value ≤ 0.05).

### 4.6. Quantitative Real-Time PCR (qRT-PCR)

The cuttage seedlings of *R. chinensis* ‘Old Blush’ and the grafted seedlings of *R. beggeriana* were used for qRT-PCR, and the cold treatment of the two species is the same as in Section 2.6. Total RNA was extracted using the Quick RNA Isolation Kit (Huayueyang Biotech, Beijing, China). The first-strand cDNA was synthesized using the PrimeScript™ RT reagent Kit with gDNA Eraser (Takara Bio Inc., Dalian, Liaoning Province, China). The qRT-PCR reactions were performed using TB Green™ Premix Ex Taq™ II (TaKaRa, Japan) and carried out on a CFX96 Real-Time PCR System (Bio-Rad Laboratories, Hercules, CA, USA). Conditions for the reaction were as follows: 95 °C for 30 s, followed by 40 cycles of 95 °C for 5 s and 60 °C for 30 s. The specific primers for *SWEET* genes were designed using the Real-time PCR (TaqMan) Primer and Probes Design Tool (https://www.genscript.com/tools/real-time-pcr-taqman-primer-design-tooland, accessed on 18 December 2022) and have been listed in Appendix A. *Rba-Tubulin* was used as the reference gene. All reactions were performed in three biological replicates, and the 2^−ΔΔCt^ method was applied to calculate the relative expression. GraphPad Prism 8.4.0 (https://www.graphpad.com, accessed on 18 December 2022) software was used to generate graphs and to perform statistical analyses. A *p*-value of <0.05 was considered to determine the significance level of the data.

### 4.7. Subcellular Localization in Tobacco

The pCAMBIA1300-c-GFP vector was used for the subcellular localization of the *RbSWEET* genes in *Nicotiana benthamiana.* The CDS sequences of three *RbSWEET* genes were amplified using a forward primer containing a HindIII restriction site and a reverse primer containing a KpnI restriction site. The amplification products were recovered using the FastPure Gel DNA Extraction Mini Kit (Vazyme Bio Co., Nanjing, Jiangsu Province, China) and then recombined with the digested plasmid by using the ClonExpress^®^ Ultra One Step Cloning Kit (Vazyme). The recombined plasmids were later transformed into *Agrobacterium tumefaciens* strain GV3101. The infection solution was prepared with sterile water containing 10 mM MgCl_2_, 10 mM 2-Morpholinoethanesulfonic Acid (MES), and 0.1 M Acetosyringone (AS), and then the *A. tumefaciens* was resuspended to an optical density (OD600) of 1.0. After standing in the dark for 3 h, the infection solution was injected into the leaf of *N. benthamiana* with a syringe. Two days after infiltration, the transient expression position of the *RbSWEET* proteins was observed by a Leica TCS SP8 fluorescence microscope (Leica Microsystems, Inc., Buffalo Grove, IL, USA) using filter blocks to select for spectral emission at 488 nm, and the empty vector was used as a control.

## 5. Conclusions

In this study, 25 *SWEET* genes were firstly identified and analyzed in *R. chinensis* ‘Old Blush’. Gene structure and motif analysis indicated that members of the SWEET family in the rose are highly conserved among the species. The prediction of *Cis*-acting elements suggested that some family members may be involved in complex signaling pathway regulation. The transcriptome data showed that there were at least eight DEGs of *SWEETs* in the leaves and shoots of *R. chinensis* ‘Old Blush’ and *R. beggeriana* after cold treatment. The qRT-PCR results further disclosed that there were various expression patterns of *SWEET* genes in two species. *SWEET1* and *SWEET4b* were relatively conserved in *Rosa* L., with similar increased trends both in tissues and species. However, *SWEET2a* and *SWEET10c* exhibited the species difference after cold treatment that sharply upregulated in the leaves of the deciduous and extremely cold-tolerant species *R. beggeriana* but not the evergreen and moderately cold-tolerant species *R. chinensis* ‘Old Blush’. The results indicated that *SWEET2a* and *SWEET10c* might be the crucial candidates to participate in the acquisition of cold tolerance in rose. Our findings provide important information for further research on the function analysis of *SWEET* genes in cold response and their application in the breeding of rose varieties with strong cold tolerance.

## Figures and Tables

**Figure 1 plants-12-01474-f001:**
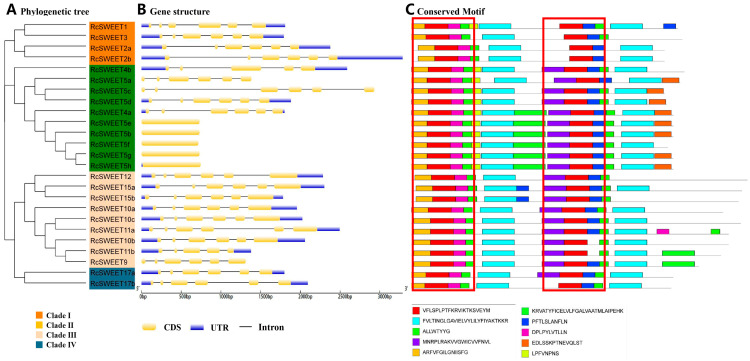
Phylogenetic relationship, gene structure, and conserved motif analysis of the *RcSWEET* genes. (**A**) Phylogenetic tree of 25 RcSWEET proteins. The unrooted neighbor-joining (NJ) phylogenetic tree was constructed with MEGA7 using amino acid sequences of 25 RcSWEET proteins, and the bootstrap test replicate was set at 1000 times. (**B**) Gene structure of *RcSWEET* genes. Yellow boxes represent CDS, and black lines represent introns. The upstream/downstream regions of the *RcSWEET* genes are indicated in blue boxes. (**C**) Distributions of conserved motifs in *RcSWEET* genes. Ten putative motifs are indicated in different colored boxes. The MtN3_slv domains were marked in two red frames.

**Figure 2 plants-12-01474-f002:**
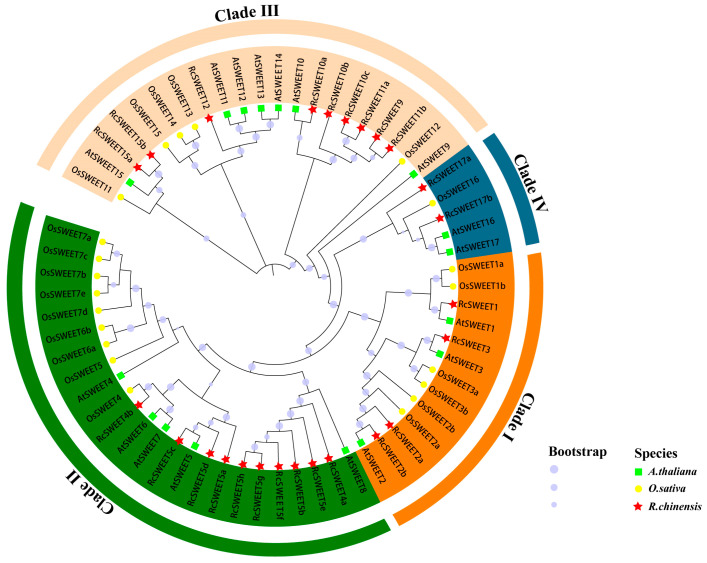
Phylogenetic tree of SWEET proteins of Arabidopsis, rice, and rose. The phylogenetic tree was constructed using the NJ (neighbor joining) method with 1000 bootstrap replications. The 4 subfamilies were distinguished by different colors.

**Figure 3 plants-12-01474-f003:**
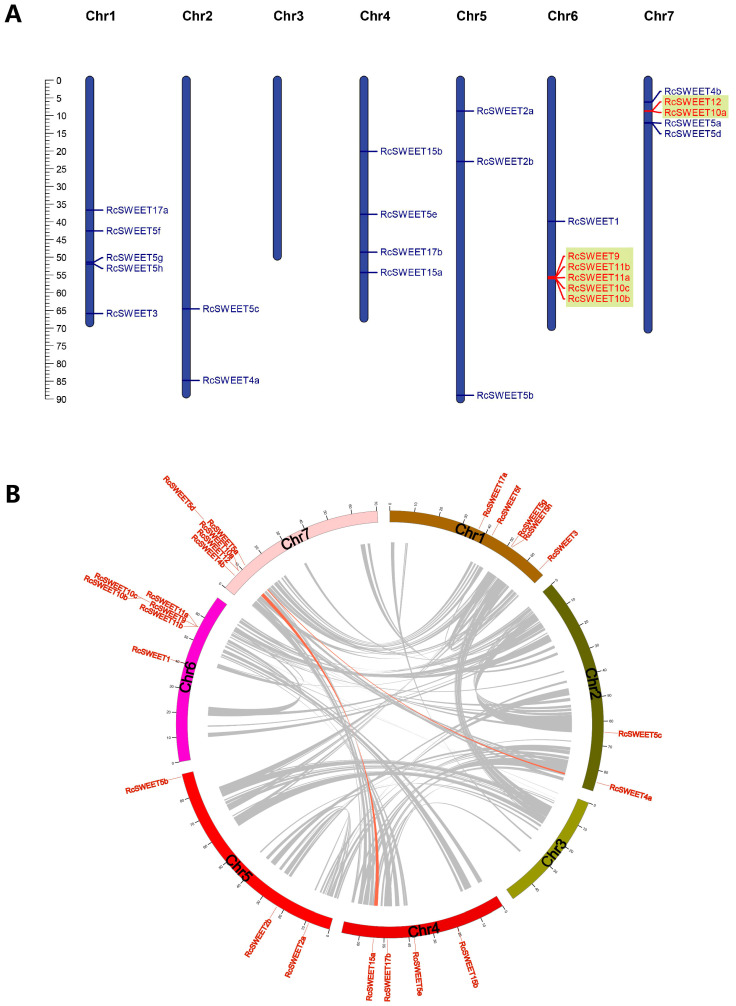
Chromosomal location and gene duplication of the *RcSWEET* genes. (**A**) Tandem duplication of *RcSWEET* genes. The tandemly duplicated genes are marked by laurel-green rectangles. (**B**) Segmental duplication of the *RcSWEET* genes. The segmentally duplicated genes of the *R. chinensis* genome are connected by gray lines, and the segmentally duplicated genes of the *RcSWEET* family are highlighted.

**Figure 4 plants-12-01474-f004:**
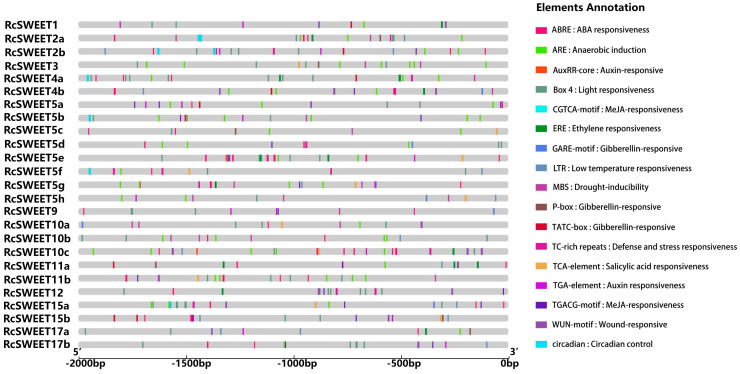
Predicted cis-elements in the promoters of the *RcSWEET* genes. Promoter sequences (−2000 bp) of 25 *RcSWEET* genes were analyzed by PlantCARE. Seventeen stress-relevant cis-acting elements are indicated in different colored boxes.

**Figure 5 plants-12-01474-f005:**
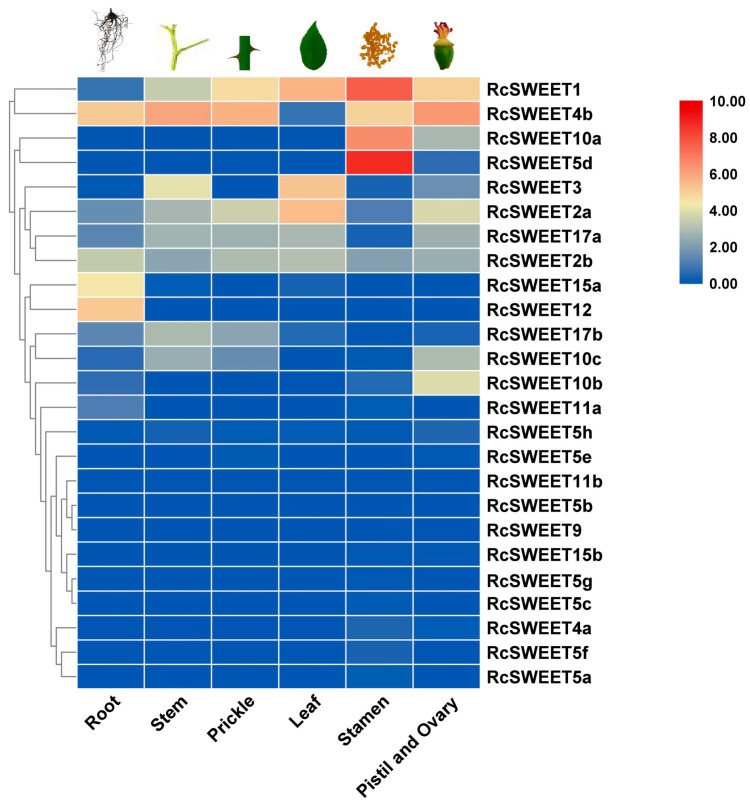
Expression patterns of *RcSWEET* genes in different tissues. Expression levels are shown as the log_2_ FPKM values obtained from the RNA-Seq data.

**Figure 6 plants-12-01474-f006:**
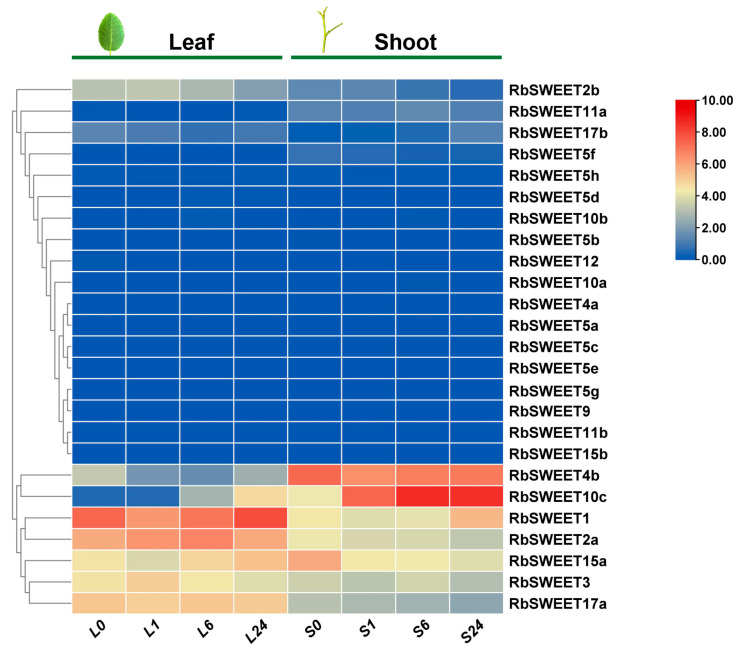
Transcriptional levels of *RbSWEET* genes after cold stress in *R. beggeriana*. Heatmap showing expression patterns of *RbSWEET* genes in the leaves and shoots treated at 4 °C for 0 h, 1 h, 6 h, and 24 h, represented by L0, L1, L6, L24, and S0, S1, S6, S24, respectively. Expression levels are shown as the log_2_ FPKM values obtained from the RNA-Seq data.

**Figure 7 plants-12-01474-f007:**
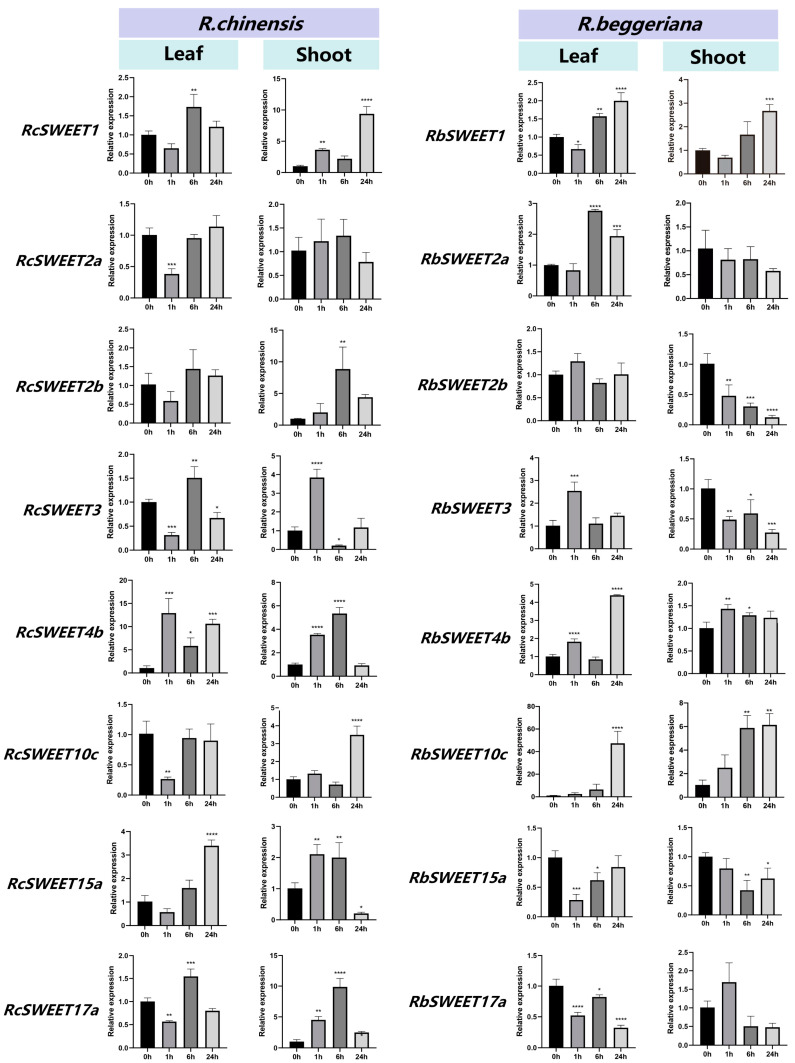
Expression levels of DEGs or highly expressed *SWEET* genes in *R. chinensis* ‘Old Blush’ and *R. beggeriana* via qRT-PCR. Error bars indicate standard deviation, and asterisks indicate significant differences between the control (0 h) and 4 °C treatment for 1 h, 6 h, 24 h, * *p* < 0.05, ** *p* < 0.01, *** *p* < 0.001, and **** *p*< 0.0001.

**Figure 8 plants-12-01474-f008:**
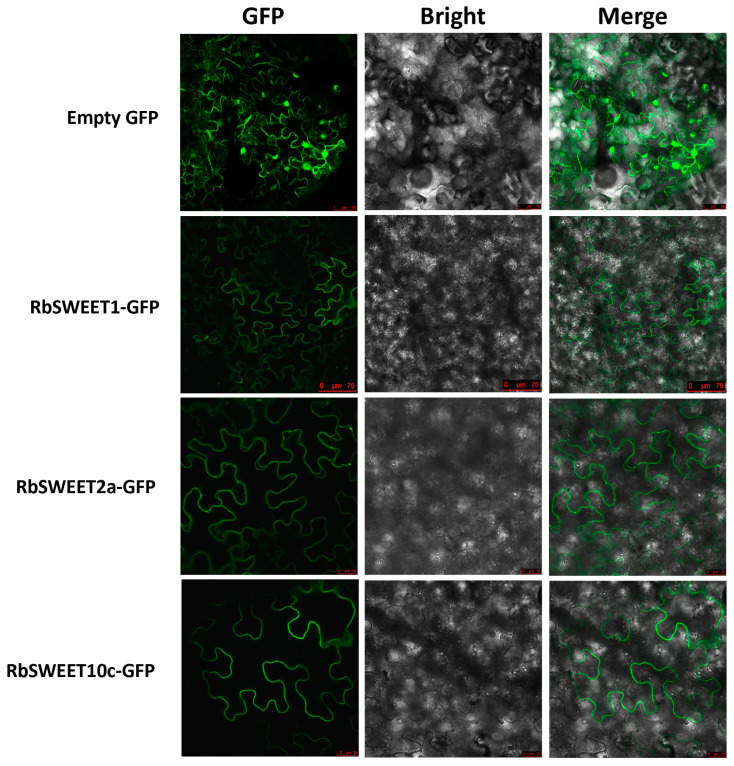
Subcellular localization of RbSWEET-GFP fusion proteins in tobacco leaves.

**Table 1 plants-12-01474-t001:** Characteristics of the *RcSWEET* gene family members in *R. chinensis* ‘Old Blush’.

Gene Name	Accession No.	Clade	Chr. No.	Location	gDNA (bp)	CDS (bp)	Protein (aa)	MW (Da)	PI
*RcSWEET1*	LOC112172425	I	6	NC_037093.1 (39918414…39920220)	1807	744	247	27,128.17	9.58
*RcSWEET2a*	LOC112201532	I	5	NC_037092.1 (8689160...8691537)	2378	708	235	26,347.62	9.57
*RcSWEET2b*	LOC112166177	I	5	NC_037092.1 (22992540…22995828)	3289	708	235	26,284.25	8.95
*RcSWEET3*	LOC112181844	I	1	NC_037088.1 (65904502…65906291)	1790	756	251	28,320.45	8.90
*RcSWEET4a*	LOC112185269	II	2	NC_037089.1 (84804695…84806493)	1799	732	243	26,958.32	9.71
*RcSWEET4b*	LOC112175327	II	7	NC_037094.1 (6173469…6176058)	2590	762	253	28,128.47	9.58
*RcSWEET5a*	LOC112177410	II	7	NC_037094.1 (12047796…12049179)	1384	750	249	28,162.82	8.92
*RcSWEET5b*	LOC112164442	II	5	NC_037092.1 (89007600…89008331)	732	732	243	27,298.40	9.12
*RcSWEET5c*	LOC112184760	II	2	NC_037089.1 (64595639…64598570)	2932	708	235	26,374.52	9.41
*RcSWEET5d*	LOC112177411	II	7	NC_037094.1 (12068260…12070141)	1882	714	237	26,541.70	9.62
*RcSWEET5e*	LOC112196855	II	4	NC_037091.1 (37884667…37885398)	732	732	243	27,172.30	8.91
*RcSWEET5f*	LOC112167162	II	1	NC_037088.1 (42566318…42567034)	717	717	238	26,881.01	8.96
*RcSWEET5g*	LOC112186521	II	1	NC_037088.1 (51365641…51366372)	732	732	243	27,261.37	8.96
*RcSWEET5h*	LOC112190416	I	1	NC_037088.1 (51946076…51946820)	745	732	243	27,273.32	8.96
*RcSWEET9*	LOC112171608	III	6	NC_037093.1 (55529800…55531112)	1313	801	266	29,697.38	9.30
*RcSWEET10a*	LOC112176691	III	7	NC_037094.1 (8669485…8671441)	1957	867	288	32,706.42	8.18
*RcSWEET10b*	LOC112174578	III	6	NC_037093.1 (55634260…55636316)	2057	882	293	32,633.41	6.51
*RcSWEET10c*	LOC112173222	III	6	NC_037093.1 (55593310…55595334)	2025	915	304	33,868.54	10.11
*RcSWEET11a*	LOC112173492	III	6	NC_037093.1 (55588084…55590577)	2494	882	293	32,435.99	9.67
*RcSWEET11b*	LOC112173793	III	6	NC_037093.1 (55578837…55580213)	1377	861	181	31,799.93	9.23
*RcSWEET12*	LOC112180477	III	7	NC_037094.1 (8655605…8657888)	2284	933	310	34,702.73	7.50
*RcSWEET15a*	LOC112196569	III	4	NC_037091.1 (54285212…54287511)	2300	918	305	34,033.00	6.51
*RcSWEET15b*	LOC112198953	III	4	NC_037091.1 (20120784…20122566)	1783	909	302	33,623.13	5.93
*RcSWEET17a*	LOC112176490	IV	1	NC_037088.1 (36716888…36718685)	1798	732	243	26,871.80	9.19
*RcSWEET17b*	LOC112200151	IV	4	NC_037091.1 (48596784…48598877)	2094	726	241	27,071.79	7.24

## Data Availability

All data are included in the main text.

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
