# Peer review of "Genome-Wide Identification of the Rose SWEET Gene Family and Their Different Expression Profiles in Cold Response between Two Rose Species"

_plants, 2023, doi:10.3390/plants12071474_

Round 1
Reviewer 1 Report
The text in the pictures should be enlarged for better readability.
Author Response
Dear Reviewer,
Thank you very much for your tireless reviewing. We revised the manuscript carefully according to your suggestions and advices. Please see them in the attached word files.
Best regards,
Prof. Dr. Shuhua YANG
No. 12 Zhongguancun South Street, Haidian District, Beijing, 100081, China
Institute of Vegetables & Flowers, Chinese Academy of Agricultural Science
Tel & Fax: 0086-10-82109542
Email: yangshuhua@caas.cn

Reviewer 2 Report
Genome-Wide Identification of Rose SWEET Gene Family and Their Different Expression Profiles in Cold Response between Two Rose Species
Comments
1. In abstract, first line is little confusing, write it again.
2. Results seems to be highly descriptive. Author may focus on major findings.
3. In Line no. 51 and 317 correct the spelling of Arabidopsis thaliana and segregation
4. In line no. 53 italicize the “AtSWEET9/Nec1” as I think here you are talking about gene.
5. Please follow the uniform pattern to write the name of plants either use biological name or common name.
6. In line no. 113 there is no need to give the abbreviation of amino acid length.
7. The figure quality of figure no. 7 should be improved.
8. In line 227 please provide the full form of DEGs as you have written its full form later in the manuscript.
9. In line no. 324 synegic word should be corrected.
10. In line no. 394 and 397 cite the references.
11. The discussion part about subcellular localization is not given, write one or two lines about its localization.
12. Since the Ms is mostly based on in silico work, author can also study its interaction analysis following the Ms https://www.mdpi.com/1422-0067/23/23/14867 which will provide the interacting partners of SWEET transporter which will highlights the coordinated functioning of this transporter with other sugar transporters.
13. Some evolutionary analysis like KA/KS, duplication events analyses would further improve the Ms. Author may follow https://www.mdpi.com/2223-7747/11/7/911 for methodology.
Author Response

(The authors gave the same response as above.)

Reviewer 3 Report
This study is focused on the characterization of the SWEET gene family in rose. The authors performed a search of such genes in Rosa chinensis genome, identifying 25 candidates which where then subjected to phylogenetic analyses, the study of regulative upstream sequences, the involvement in abiotic stress responses and expression patterns by both RNA seq and qRT-PCR in two species divergent for cold tolerance after cold treatment. Following the results of expression analyses, the authors formulated hypotheses about the different roles and expression of SWEET genes in cold tolerance; i.e. how some resulted relatively conserved among species while others are divergent.
The SWEET genes have been extensively studied for their function in a number of plant species, and their importance in the tolerance of different stresses is well documented; therefore, a study of this gene family in a commercially important genus such as Rosa can be extremely useful in the development of improved varieties.
The topic is well presented in the introduction, starting from the information in other plant taxa and giving an overview on the role of SWEET genes in a number of physiological and stress-related aspects; the experimental design is adequate and allowed a wide and exhaustive characterization of the SWEET gene family. The results are correctly supported by the data presented, although in some cases a more detailed discussion would be useful.
I would suggest some minor revisions in order to improve some aspects of the manuscript.
The results can be improved in the presentation in order to make them more comprehensible; in some parts, in my opinion, the conclusions are not exactly supported by the data presented. In some cases the results should be better explained and hypotheses should be formulated.
In general, English requires a moderate revision. Check some sentences which lack the verb.
Please find below a more detailed list of suggested revisions:
L32: I would modify as follows: “SWEET (“Sugars Will Eventually be Exported Transporter”) is a…….”
L51: Check the spelling of Arabidopsis
L70: delete “the”
L91: “the species” should be “species”
L120: The caption of Table 1 should be more informative
L131-132: in my opinion it is not correct to say that “the results suggest”. Rather, I would say that the gene structure correlated with the clustering of genes into subfamilies, according to their sequence
L171-173: please check the English
L173: I would end the sentence with reference to fig. 3A, while in L175 I would put “Figure 3B”
L194: as far as I understood, “genes” should be “elements”
L203: …… in which conditions? Please specify
L206: “applied” should be “analyzed” (unless I did not understand the meaning of the sentence)
L213: if I understood correctly, “shoots” should be “roots”
L236: This is true for 17a, but for 3 I can see a difference in expression levels following cold treatment in both leaves and shoots.
L252: differences are not significant for 2b in the figure
L253: to be more accurate, I would specify for 4b “compared to the control”, because the expression decreases after 6h of cold treatment
L257: 2b is repeated twice. Did you mean 2b and 3?
L260: “that” should be “for”
L261: the expression patterns 17a doesn’t actually look different between RNA-seq and qRT-PCR. As far as I can see from the figures, an initial decrease is followed by an increase at 6h and then again a decrease
L267: in the expression of 2a and 10c, there seems to be actually a significant decrease in expression after 1h. Please see my comment on figure 5.
L286-287: this sentence is not necessary in this place. It is more appropriate for the discussion section
L324: “synergic” should be “synergically”
L329: Are you sure that the expression patterns are different in the two genes? In both I see a much higher expression in roots compared to the other organs. Did you mean 10a and 12?
L338: “is usually occurred” should be “usually occurs”
L387: this is not true for for 4b and 10c
L391: I suggest to modify the sentence as follows: “…..also found in the two rose species studied”
L407: “However” should better be “instead”
L409: the expression pattern is a bit more complicated than this; an initial sharp decrease is followed by restoration to T0 levels and this is evident for both 2a and 10c. Do you have any hypotheses about this?
L438: “saliva” should be “sativa”
L473: why use two different materials for the two species (i.e. cottage and grafted seedlings)? Is this because of a different vigour?
L509: “conservative” should maybe be “conserved”
Figure 5: this applies to all the figures with heatmaps. I think it would be helpful if the numeric expression values used to generate the figure were added to each rectangle, as in some cases the difference in color hues are not clearly visible.
Figure 7: the whole figure should maybe be bigger. I suggest to magnify the gene names for easier reading. Also the notation A-D could be avoided, and simply substituted with “leaves” and “shoots”
Author Response

(The authors gave the same response as above.)

Round 2
Reviewer 2 Report
Ms is improved. It can be accepted now.